# Are There Differences in Inflammatory and Fibrotic Pathways between IPAF, CTD-ILDs, and IIPs? A Single-Center Pilot Study

**DOI:** 10.3390/ijms232315205

**Published:** 2022-12-02

**Authors:** Patrycja Rzepka-Wrona, Szymon Skoczyński, Adam Barczyk

**Affiliations:** Department of Pneumonology, Faculty of Medical Sciences in Katowice, Medical University of Silesia, 40-055 Katowice, Poland

**Keywords:** interstitial pneumonia, bronchoalveolar lavage, interstitial lung disease, IPAF, CTD-ILDs, IIPs

## Abstract

In this pilot study, we aim to determine differences in pathogenetic pathways between interstitial pneumonia with autoimmune features (IPAF), connective-tissue-disease-associated interstitial lung diseases (CTD-ILDs), and idiopathic interstitial pneumonias (IIPs). Forty participants were recruited: 9 with IPAF, 15 with CTD-ILDs, and 16 with IIPs. Concentration of transforming growth factor beta (TGF-β1), surfactant proteins A and D (SP-A, SP-D), interleukin 8 (IL-8), and chemokine 1 (CXCL1) were assessed with ELISA assay in bronchoalveolar lavage (BAL) fluid. We revealed that IL-8 and TGF-β1 concentrations were significantly lower in the IPAF group than in the CTD-ILD group (*p* = 0.008 and *p* = 0.019, respectively), but similar to the concentrations in the IIP group. There were significant correlations of IL-8 (rs = 0.46; *p* = 0.003) and CXCL1 (rs = 0.52; *p* = 0.001) and BAL total cell count (TCC). A multivariate regression model revealed that IL-8 (β = 0.32; *p* = 0.037) and CXCL1 (β = 0.45; *p* = 0.004) are significant predictors of BAL TCC. We revealed that IL-8 and TGF-β1 BAL concentrations vary in patients with different ILDs and found that IL-8 is a predictor of BAL TCC in IPAF. However, this needs to be confirmed in a multicenter cooperative study (ClinicalTrials.gov Identifier: NCT03870828).

## 1. Introduction

More than 200 distinct entities are grouped under the heading of interstitial lung disease (ILD). Among them, the most recognized are idiopathic interstitial pneumonias (IIPs), connective-tissue-disease-associated interstitial lung diseases (CTD-ILD), and interstitial pneumonia with autoimmune features (IPAF).

IPAF is a relatively novel entity, defined by the European Respiratory Society (ERS) and American Thoracic Society (ATS) Work group as an ILD with features of autoimmunity [1].

Currently, the target of biomarker development in this field has been to establish measurable, comparable variables for refined classification of ILDs, prediction of prognosis, or response to treatment.

Until now, most studies have focused on serum biomarkers; however, bronchoalveolar lavage (BAL) fluid reflects local processes of inflammation and fibrosis. It is known that higher levels of IL-8 are observed in systemic-sclerosis-associated ILD (SSc-ILD) than ILD-negative Ssc. Elevated IL-8 BAL fluid concentration is also correlated with lower diffusion lung capacity for carbon monoxide (DLCO), forced vital capacity (FVC), and total lung capacity (TLC) predicted values. Higher levels also correlate with progressive fibrosis or end-stage ILD at 38 months [2].

BAL levels of TGF-β1 decreased significantly after 3 months of cyclophosphamide treatment in patients with Sjogren’s syndrome [3]. Higher concentrations correlate with the extent of interstitial abnormalities (high-resolution computed tomography; HRCT score) in IIPs [4].

To the best of our knowledge, there are no prospective studies concentrating on the development of disease biomarkers, delineating distinct pathological pathways and comparing clinical outcomes between patients with IPAF and CTD-ILDs or IIPs.

Therefore, many questions regarding IPAF remain unanswered: Will it transform into a definite CTD-ILD? Can we identify prognostic and/or diagnostic biomarkers of IPAF? How do patient-related outcomes (e.g., survival, mortality, frequency of acute exacerbations (AEs)) differ from those in CTD-ILDs or IIPs? What are the differences in radiological, serological, functional, and clinical parameters between individuals with IPAF, CTD-ILD, and IIP? What are the most potent prognostic factors in IPAF? How should we approach therapy in the case of progressive fibrosing ILD (PF-ILD) in the course of IPAF? What are the mechanisms of interstitial lung fibrosis in IPAF, and are there common pathways with CTD-ILD and IIPs?

To answer these questions, we have initiated a multicenter prospective study. We plan to compare the concentration of proteins in BAL fluid between three study groups, IPAF, CTD-ILD, and IIPs, and to characterize and compare local inflammation-fibrosis pathways in these three different entities. In order to determine the feasibility of this study, we decided to conduct a single-center preliminary study.

In our opinion, this pilot study will provide us with background for expanding and modifying the choice of candidate biomarkers. We are looking forward to prospective cooperation with other pulmonological and rheumatological centers via the corresponding author.

## 2. Results

### 2.1. Study Group Characteristics

In the IIP group, the percentage of males was significantly higher, which represents the typical sex distribution in analyzed disease groups. There were no other statistically significant differences between groups. Prevalence of symptoms of gastrointestinal reflux disease (GERD) was significantly higher in the CTD-ILD group than in the IPAF group (78.6% vs. 11.1%). No statistically significant differences were revealed in symptoms’ duration.

Radiological pattern of usual interstitial pneumonia (UIP) was significantly more frequently in the IIP group (62.5% vs. 20% in the CTD-ILD group and 0% in the IPAF group).

No statistically significant differences were revealed between BAL parameters; however, we have revealed important abnormalities in the BAL result. Below, we present examples of HRCT imaging for each study group. Figure 1 presents radiological pattern of UIP in course of IPF. Figure 2 is an example of UIP pattern in course of SSc-ILD, whereas Figure 3 shows NSIP in a patient with IPAF.

We present detailed study group characteristics in Table 1. In Table 2 we present patient comparison in terms of BAL fluid parameters.

### 2.2. Comparative Analysis in Terms of Testing for Autoantibodies’ Presence and Titer

In Table 3, we present results of group comparative analysis in terms of autoantibodies’ presence and titer. For the RF, a threshold of ≥14 IU/mL was regarded as a positive result. For ANA, titer 1:320 was considered positive. Regardless of titer, ANA with either a nucleolar or centromere-staining pattern were included as an IPAF criterion. For anti-cyclic citrullinated peptide antibodies (anti-CCP), a concentration > 20 U/mL was regarded positive.

Statistical analysis of the study groups did not reveal significant differences regarding the presence of RF anti-CCP.

Overall, 12.5–33.3% of patients tested positive for RF (≥14 IU/mL), and 6.3–50% of patients tested positive for anti-CCP. The lowest prevalence of anti-CCP was reported in the IIP group. There were statistically significant differences between the groups regarding ANA prevalence.

The percentage of patients who tested negative for ANA in the IIP group was significantly higher than in the IPAF and CTD-ILD group (81.3% vs. 11.1% and 33.3%, respectively).

### 2.3. Group Membership and Cytokine Levels

To test whether group membership is differentiated by levels of IL-8, TGF-β1, SP-D, SP-A, and CXCL1, general linear model analysis was performed. In case of IL-8, TGF-β, SP-A and CXCL1, log gamma distribution was used. For SP-D, generalized linear model with Gauss distribution and log link function was performed. The reference group was patients with IPAF in every model.

#### 2.3.1. IL-8

The analysis revealed that the model was well-fitted to the data (χ^2^(2) = 9.10; *p* = 0.011; AIC = 398.14). Parameters in the model revealed that in the IPAF group, IL-8 concentration was significantly lower than in the CTD-ILD group by 1.14 unit. In addition, for the estimated marginal means, a post hoc analysis was conducted using the Bonferroni test for pairwise comparisons, which revealed that IL-8 concentration was significantly higher than in the IPAF (*p* = 0.008; Figure 4).

#### 2.3.2. TGF-β1

The analysis revealed that the model was well-fitted with the data (χ^2^(2) = 8.40; *p* = 0.015; AIC = 613.60). Parametric analysis in the model revealed that TGF-β1 concentration was significantly higher by 0.81 unit in the CTD-ILD group than in the IPAF group. Furthermore, for the estimated marginal means, a post hoc analysis was conducted using the Bonferroni test for pairwise comparisons, which revealed that in the CTD-ILD group, TGF-β1 concentration was significantly higher than in the IPAF or IIP groups (*p* = 0.019; Figure 5).

#### 2.3.3. SP-A, SP-D

There were no statistically significant differences or trends between groups regarding BAL concentrations of SP-A and SP-D.

#### 2.3.4. CXCL1

The analysis revealed that the model was well-fitted with the data (χ^2^(2) = 6.0; *p* = 0.041; AIC = 613.02). Post hoc analysis did not reveal any significant differences in CXCL1 concentration between the groups. Post hoc analysis did not reveal significant differences between the groups in SPA concentration. The analysis revealed that in the CTD-ILD group, CXCL1 concentration was higher by 0.64 unit than in the IPAF group. Post hoc analysis for estimated marginal means did not reveal significant differences between the groups.

### 2.4. Relationship between BAL Protein Concentration and ILD Radiological Pattern

No statistically significant correlations or trends were found between BAL protein concentration and HRCT patterns.

### 2.5. Relationship between BAL Protein Concentration and Symptom Duration

No statistically significant correlations or trends were found between BAL protein concentration and symptom duration.

### 2.6. Correlation between BAL Protein Concentration and Pulmonological Tests Parameters

Table 4 presents correlation coefficients for BAL protein concentration and pulmonological test parameters. Analysis revealed a moderate positive correlation between IL-8 concentration and BAL TCC (rs = 0.46; *p* = 0.003), a moderate negative correlation between CXCL1 concentration and DLCO (Eta = −0.36; *p* = 0.024), and a strong positive correlation between CXCL1 concentration and BAL TCC (rs = 0.52; *p* = 0.001).

Table 5 presents the correlation between BAL protein presence and autoantibodies presence. The analysis revealed a moderate negative correlation between TGF-β1 concentration and ANA presence (Eta = −0.35; *p* = 0.025).

### 2.7. Correlation between BAL Protein Concentration and Rheumatological Symptoms

The statistical analysis has revealed a moderate positive correlation (Eta = 0.33; *p* = 0.037) between IL-8 BAL concentration and teleangiectasias presence and a moderate positive correlation between CXCL-1 concentration and muscle weakness (Eta = 0.37; *p* = 0.022).

In order to determine whether cytokine concentrations were BAL TCC predictors, multiple linear regression analysis was performed, where predictors were the five analyzed cytokines, and BAL TCC was the explanatory variable.

It was revealed that IL-8 and CXCL-1 were significant predictors of BAL TCC (β = 0.32; *p* = 0.037 and β = 0.45; *p* = 0.004, respectively).

The analysis revealed that the model was well-fitted to the data (F(5.31) = 3.75; *p* = 0.009) and explained, in total, 27.7% of the variances of BAL TCC. Regression coefficients for this model are presented below in Table 6.

## 3. Discussion

The major finding revealed in our pilot study is the presence of a statistically significant difference in IL-8 BAL concentration between IPAF and CTD-ILD groups, namely, IL-8 concentration in the IPAF group is significantly lower than in the CTD-ILD group, and similar to the concentration in the IIP group.

The statistical analysis has also revealed a moderate positive correlation between IL-8 BAL concentration and teleangiectasias presence.

IL-8 is secreted by various cells, including fibroblasts, endothelium, mesothelium, monocytes, neutrophils, and malignant tumor cells. It is released as a response to inflammatory stimuli. It also plays an important role in wound healing [7], and its role is to recruit lymphocytes, T cells, and other inflammatory cells into inflammation areas by activating neutrophils [8]. Moreover, IL-8 attracts fibroblasts and stimulates their migration and deposition of extracellular cell matrix (ECM) proteins in the process of wound healing in vivo [9].

IL-8 was proven to be an angiogenic factor as early as 1992 [10] based on studies of malignant neoplasms. Kitadai et al. [11] revealed high concentrations of IL-8 in gastric carcinoma specimens in comparison with normal mucosa specimens. Furthermore, IL-8 concentrations correlated positively with the extent of vascularity [11].

Yalçınkaya et al. [12] correlated levels of vascular biomarkers with nailfold videocapillaroscopy findings in patients with SSc. The authors classified the results according to Cutolo classification [13] as early, active, and late pattern. Lower IL-8 serum concentrations were revealed in patients with the early pattern, which suggests that IL-8 plays a role in the active or late phase of vascular injury.

The drawback of our pilot study was, unfortunately, lack of access to videocapillaroscopy. However, in a study on 33 patients with SSc, Pizzorni et al. [14] investigated potential correlations between nailfold microangiopathy severity and skin teleangiectasia patterns. It was revealed that the severity of capillaroscopic findings correlated with the total number of skin teleangiectasias.

Therefore, we assume that increased IL-8 BAL concentration may be a biomarker of vascular injury in CTD-ILD patients.

The statistical analysis also revealed a moderate positive correlation between BAL IL-8 concentrations and BAL total cell count.

IL-8 is, among its other functions, a neutrophil chemotactic factor, expressed in neutrophils and alveolar macrophages [15]. IL-8 binds to its G-protein-coupled CXC chemokine receptors, CXCR1 and CXCR2, which activates a phosphorylation cascade to stimulate neutrophil activation and chemotaxis [16].

The role of IL-8 in alveolar inflammation was explored in the context of acute respiratory distress syndrome (ARDS) pathogenesis. BAL IL-8 was proposed as a prognostic biomarker in predicting the occurrence and outcomes for ARDS not associated with infection of the novel coronavirus. Elevated IL-8 BAL levels show significant correlation with mortality in pneumonia, sepsis, and non-specific ARDS [17].

Associations between BAL cell profiles and cytokine levels in ILD have been studied by many authors, revealing higher BAL TCC and BAL neutrophil count in IIPs [18].

The authors measured BAL TCC in patients with diffuse panbronchiolitis and IPF vs. healthy volunteers. The mean TCC and mean neutrophil counts were significantly higher in patients with interstitial pneumonia than in healthy controls. They also revealed a correlation between BAL IL-8 concentrations and neutrophil count in BAL. It is hypothesized that alveolar macrophages are the source of IL-8 in the BAL fluid of IPF patients. Carre et al. [19] revealed increased expression of IL-8 gene in these cells by reverse transcription polymerase chain reaction. They also correlated levels of IL-8 mRNA with BAL neutrophil count.

In our study, no association was revealed between BAL neutrophil count or mononuclear cell count and IL-8; however, we may obtain different results in the full-scale study, where we plan to recruit 240 individuals. Moreover, in spite of the absence of significant between-group differences, neutrophil count was outside the normal range in all groups and the lymphocyte count exceeded the upper limit of normal in CTD-ILD group and was below the norm in IIP group. At the same time, the lymphocyte count was within predicted values for IPAF patients. Therefore, we hypothesize that in IPAF, CTD-ILD, and IIP pathogenesis, there are significant differences in both cytokine and cellular pathways.

Liang et al. [20] in a study on the association of CXCL1 with IPAF, revealed that, in comparison with idiopathic interstitial lung diseases, increased plasma levels of CXCL1 were exhibited in individuals with IPAF. Moreover, there was also an association between increased CXCL1 plasma levels and DLCO. The highest concentrations were found in IPAF patients experiencing an acute exacerbation. This is in concordance with our analyses, which revealed higher CXCL1 concentrations in subjects with DLCO < 60% predictive values.

Remarkably, in the study by Liang et al., there was also a positive correlation between BAL neutrophil count in IPAF and increased CXCL1 concentrations. In our analysis, there was no correlation between CXCL1 and neutrophil count; however, a positive correlation between CXCL1 and BAL TCC was revealed.

The analysis revealed differences in CXCL1 concentrations in the groups. In the post hoc analysis of our data, however, there was no significant difference between groups in CXCL1 concentration. In the CTD-ILD group, CXCL1 BAL concentrations were higher per 0.61 unit than in the IPAF group, and in IPF group, CXCL1 BAL concentrations were greater per 0.64 unit than in the IPAF group.

Perhaps, in a larger sample size, the differences between CXCL1 levels in IPAF vs. CTD-ILD could become more apparent, especially given that parametrical analysis revealed differences between the groups.

Further statistical analysis of our study groups has revealed a moderate negative correlation between TGF-β1 BAL concentrations and ANA presence (Eta = −0.35; *p* = 0.025).

In terms of function, TGF-β1 is a protein responsible for regulating various cellular processes such as alveolar epithelial cell differentiation, fibroblast activation, and extracellular matrix metabolic processes. These processes are associated with interstitial tissue remodeling in progressive fibrosis. The key step of pulmonary fibrosis development is the differentiation of fibroblasts into myofibroblasts, which produce proteins of extracellular matrix.

TGF-β1 is a potent stimulus for myofibroblast differentiation, and its enhanced expression, mostly by alveolar macrophages and type II epithelial cells [21], has been described in a fibrotic lung. Moreover, TGF-β1 induces other particles, including integrins and matrix metalloproteinases, which contribute to further tissue remodeling and promotes interstitial fibrosis by suppressing production of anti-fibrotic agents (e.g., hepatocyte growth factor and prostaglandin E2) [22].

Patients with negative ANA are individuals with IIP, mostly IPF and iNSIP. iNSIP, according to some authors [23], is a progressive condition with poor outcomes. Hiwatari et al. [24] revealed that high levels of TGF-β1 in IPF patients may contribute to the progression of interstitial fibrosis and shorter survival.

Moreover, other authors revealed that in IPF patients, TGF-β1 levels correlate with mortality [25].

Khalil et al. [5] reported enhanced levels of TGF-β1 in localizations with active fibrosis in IPF individuals. These findings have been corroborated in a study on patients with idiopathic interstitial pneumonia (including IPF and iNSIP) and granulomatous diseases [26], which revealed a trend towards higher concentrations in the idiopathic pneumonia subgroup. Enhanced concentrations of TGF-β1 in IPF patients were found not only in lung biopsy specimens, but also in BAL fluid, which may be explained by the protein’s continuous mRNA expression in alveolar macrophages [26]. However, Meloni et al. [27] revealed no statistically significant differences in TGF-β1 concentrations in subjects with SSc-ILD, IPF, and sarcoidosis compared with healthy volunteers.

Therefore, we hypothesize that TGF-β1 may be a prognostic marker, and its enhanced levels in patients with autoimmune background may be associated with progressive fibrosis and shorter survival. However, this hypothesis should be tested in large-scale prospective studies.

### Study Limitations

There are several limitations to our study, the most important of these being the small sample size, the fact that this was a single-center study, and the fact that we had no access to videocapillaroscopy. However, we are planning to start recruitment to our multi-center study, with an estimated sample size at 240 participants. At the moment, there are eight Polish centers preparing for recruitment initiation. We are also planning for serial studies on patients to reveal relationships of BAL biomarkers and clinical, functional, and radiological progression. Moreover, to increase the value of our research, we are looking forward to international cooperation. More information is available from Patrycja Rzepka-Wrona (e-mail: patrycja.rzepka2@gmail.com).

## 4. Material and Methods

### 4.1. Study Overview

The study was conducted according to the guidelines of the Declaration of Helsinki and approved by the Medical University of Silesia Bioethics Committee (approval number KNW/0022/KB1/130/18/19, date of approval 8 January 2019). It is a preliminary study of “Clinical Characteristics of Interstitial Pneumonia with Autoimmune Features (IPAF)—A Multicenter Prospective Study” (ClinicalTrials.gov Identifier: NCT03870828).

During assessment, patients underwent pulmonary function tests (PFTs), bronchofiberoscopy (FOB) with BAL and 6-min walk test (6MWT), blood sampling, capillary blood gas analysis (CBG), and HRCT. These procedures constitute part of a standard ILD diagnostic process. TLC, FVC, forced expiratory volume in 1 s (FEV1), and DLCO were measured according to ATS/ERS guidelines [28,29,30]. Protein concentration analysis in BAL supernatant was performed by sandwich ELISA method according to the manufacturers’ protocols.

The inclusion criteria were age ≥ 18 years and a new diagnosis of IPAF, CTD-ILD, or IIP.

The exclusion criteria were: treatment with systemic glucocorticosteroids for up to 12 months prior to recruitment, respiratory tract infection 4 weeks prior to the diagnostic process, malignancy, asthma, tuberculosis, asbestosis, chronic hypersensitivity pneumonia, Hermansky–Pudlak syndrome and other causes of genetic ILD (e.g., lymphangioleiomyomatosis, tuberous sclerosis complex (TSC), Birt–Hogg–Dubé syndrome, dyskeratosis congenita), immunodeficiency syndromes (e.g., human immunodeficiency virus (HIV) seropositivity), pregnancy, and/or nursing.

Informed consent was obtained from all participants. In total, for the purpose of the pilot study, 40 patients were recruited: 9 with IPAF, 16 with IIP, and 15 with CTD-ILD. Patients were diagnosed with IPAF according to the 2015 ERS/ATS research statement [1]. IIP diagnosis was based on clinical, radiological, and/or pathological data based on the Official ATS/ERS/JRS/ALAT Clinical Practice Guideline [31]. CTDs were diagnosed by experienced rheumatologists according to the European League Against Rheumatism/American College of Rheumatology (EULAR/ACR) classification criteria [32,33,34,35].

PFTs were performed with the use of the diagnostic system MasterLab Jeager (Wuerzburg, Germany). All tests were carried out according to the ERS/ATS technical statements [28,29,30]. Finally, 6MWT was performed according to the ATS guidelines in a 30 m long corridor [36].

#### 4.1.1. Serological Testing

Antinuclear antibodies’ (ANA) presence and titer were measured by indirect immunofluorescence according to manufacturer’s instructions. ANA patterns were described according to the current International Consensus on ANA Patterns (ICAP) [37]. Rhematoid factor (RF) concentration was measured by ELISA assay according to manufacturer’s instructions.

#### 4.1.2. Radiological Testing

HRCT (≤1.5-mm slice thickness) was performed during hospitalization or was provided by the patient from up to 6 months prior to the recruitment (e.g., from an outpatient clinic) and was assessed by an independent radiologist, who was not involved in the study. Radiological analysis was performed with the use of OsiriXLite DICOM Viewer (Geneva, Switzerland). Volume imaging with slice thickness ranging from 0.625 to 1.0 mm was used. High-spatial-frequency or sharpening algorithm were used during postprocessing.

Radiological pattern of usual interstitial pneumonia (UIP) was diagnosed based on respiratory medicine societies’ guidelines [31].

#### 4.1.3. FOB and BAL

During FOB, BAL was performed according to ERS and Polish Respiratory Society guidelines [38,39]. BAL site was based on HRCT, and areas with the most evident involvement of the interstitial tissue were chosen. During FOB, the bronchofiberoscope was placed in a wedge position within the selected segment. Then, 0.9% saline with a total volume of 200 mL (at 37 degrees Celsius) was instilled through the bronchoscope. This volume was divided into aliquots per 25 mL. After the instillation of each aliquot, the saline was retrieved. The procedure was regarded as diagnostic if the lavage salvage was at least 60%.

BAL fluid was filtered through double-layer gauze to dispose of excess mucus. Differential cell count and assessment of cell viability were performed directly after filtration in a sample of well-mixed BAL fluid. Total cell count was obtained with the use of a hemocytometer, and cell viability was determined by Trypan blue exclusion. Differential cell counts were performed via cytocentrifugation with May–Grunwald–Giemsa staining and enumeration of at least 400 cells. Samples were stored in −80 degrees Celsius in a mechanical freezer.

#### 4.1.4. Protein Concentration Measurement in BAL

ELISA sandwich enzyme immunoassay was used for the quantitative measurement of candidate fibrosis biomarkers in BAL. Protein concentration was analyzed in undiluted or 1:20 diluted BAL supernatant specimens. Each BAL sample was tested twice. Below we present names of the assays and their detection limits: Interleukin 8 (IL-8) ELISA kit RD194558200R, BioVendor-Laboratorní medicína a.s. (Brno, Czech Republic): 12 pg/mL; Human TGF-β1 ELISA kit 650.010.096, Diaclone SAS (Besançon, France): 8.6 pg/mL; Human Surfactant Protein D ELISA kit RD194059101, BioVendor-Laboratorní medicína a.s. (Brno, Czech Republic): 0.1 ng/mL; Surfactant-Associated Protein A (SPA) ELISA kit SEA890Hu, Cloud-Clone Corp (Wuhan, China): 46.88 pg/mL; Enzyme-Linked Immunosorbent Assay Kit Chemokine (C-X-C motif) Ligand 1 (CXCL1) SEA041Hu, Cloud-Clone Corp (Wuhan, China): 6.1 pg/mL.

#### 4.1.5. Statistical Analysis—Overview

Statistical analysis was performed with the use of IBM SPSS Statistics 28.0 (Armonk, NY, USA) and Jamovi 2.3.12.0 (Sydney, Australia) systems. The significance level α = 0.05 was adopted for the analysis. First, analysis of basic descriptive statistics was performed with Shapiro–Wilk normality test.

In order to compare the groups in terms of nominal data, chi-square test or exact Fisher–Freeman–Halton test was performed (when expected sample size was less than 5). To compare the groups in terms of quantitative variables, Kruskal–Wallis H test was performed, and as post hoc analysis, Dunn test with Bonferroni correction of significance level was performed.

In order to determine the group differentiating role for IL-8, SP-D, SP-A, CXCL1, and TGF-β1, analysis of generalized linear models was performed. In order to determine the statistical relationships between IL-8, SP-D, SP-A, CXCL1, TGF-β1, and sociodemographic variables, pulmonological tests parameters, and clinical symptoms, Spearman correlation rank test was performed for quantitative variables, and Eta correlation coefficient test was performed to determine the statistical relationship between quantitative and nominal variables.

## 5. Conclusions

Based on our early findings, we assume that IPAF development is associated with different local inflammatory and fibrotic pathways than CTD-ILDs and IIPs.

Further studies on interleukins’ role in different stages of pulmonary fibrosis would provide new perspectives on both the pathogenesis mechanism, as well as the therapeutic strategy and drug development.

## Figures and Tables

**Figure 1 ijms-23-15205-f001:**
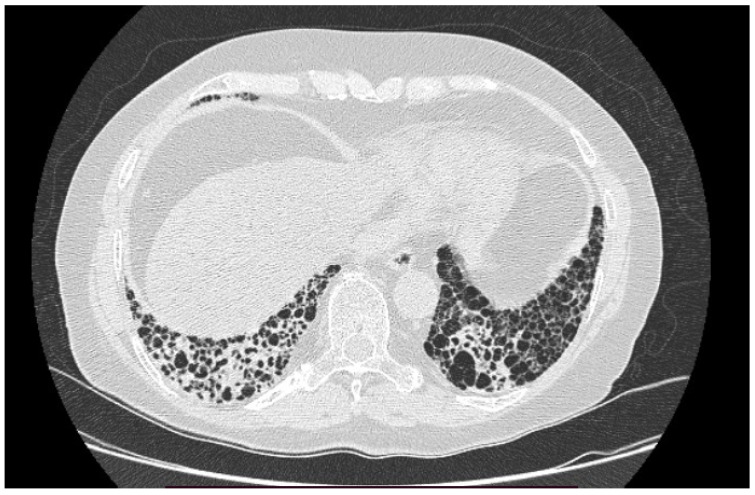
UIP in course of IPF.

**Figure 2 ijms-23-15205-f002:**
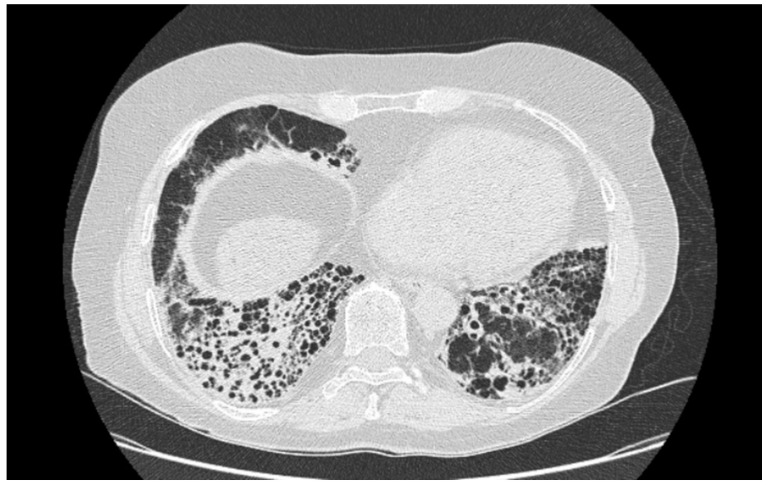
UIP in course of SSc-ILD.

**Figure 3 ijms-23-15205-f003:**
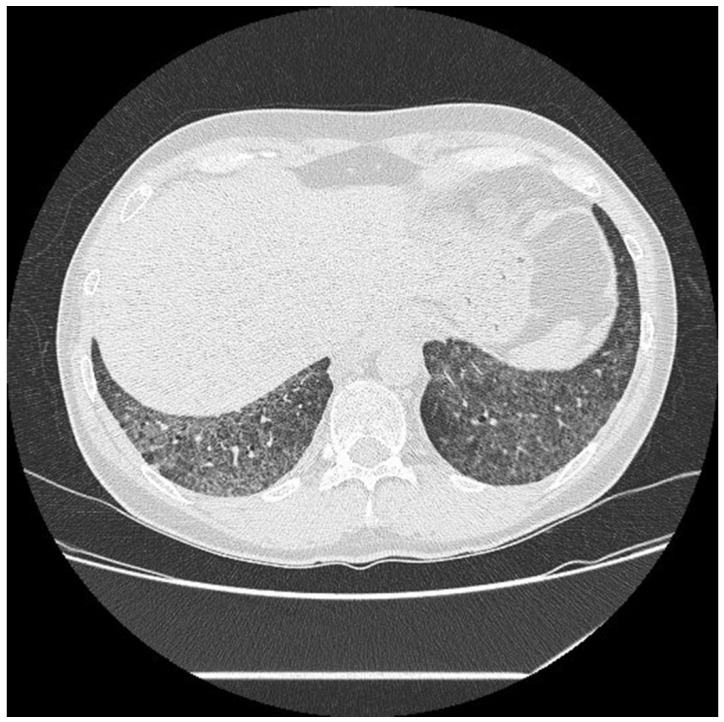
NSIP in course of IPAF.

**Figure 4 ijms-23-15205-f004:**
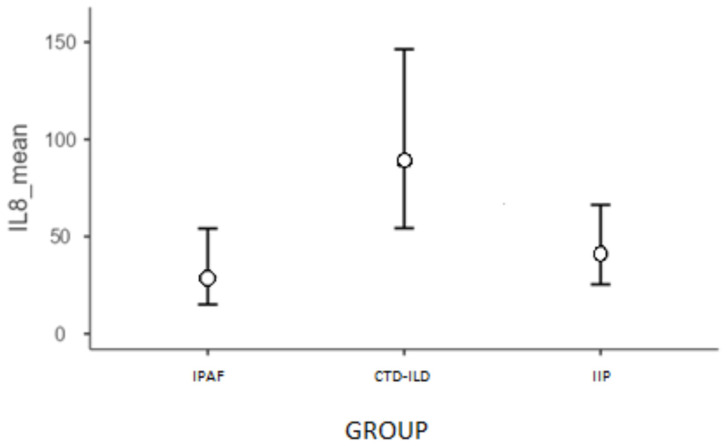
Estimated marginal means for IL-8 by group membership.

**Figure 5 ijms-23-15205-f005:**
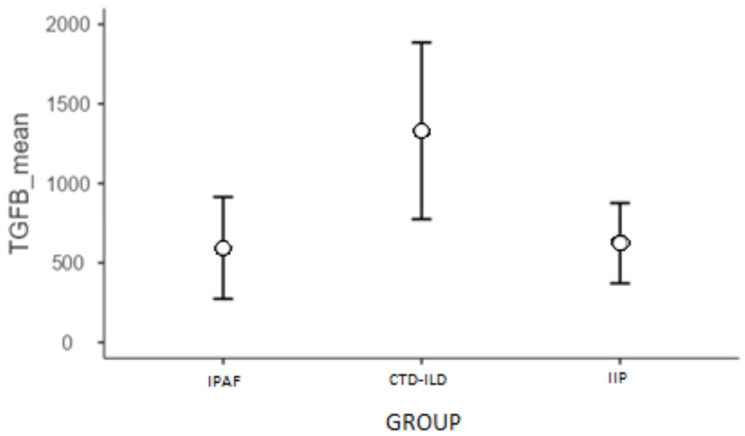
Estimated marginal means for TGF-β1 by group membership.

**Table 1 ijms-23-15205-t001:** Study group characteristics.

		IPAF		CTD-ILD		IIP		*p*
**DEMOGRAPHICS**	Age (years)	68.78 ± 7.89		65.80 ± 8.98		69.13 ± 12.21		0.43
	Sex (% female)	88.9		80.0		31.3		0.004
	Height (cm)	160.83 ± 7.08		164.80 ± 6.80		168.75 ± 8.95		0.06
	Weight (kg)	75.67 ± 11.56		72.60 ± 13.82		80.53 ± 14.52		0.51
	History of tobacco smoke exposure (%)	55.6		78.6		62.5		0.47
**FUNCTIONAL PARAMETERS**		Me	IQR	Me	IQR	Me	IQR	
	FEV_1_ (l)	1.90	0.75	1.84	0.68	2.33	1.13	0.23
	FEV_1_ (%)	83	0.15	0.77	0.29	0.85	0.25	0.53
	FVC (l)	2.74	1.50	2.45	1.60	2.84	2.08	0.54
	FVC (%)	74.00	30.00	73.00	44.00	79.00	28.00	0.54
	FEV_1_/FVC	79.55	11.91	79.20	10.36	82.40	21.16	0.467
	TLC (l)	3.92	1.32	4.89	1.86	4.55	2.41	0.28
	TLC (%)	77.00	20.50	87.00	32.00	82.50	27.50	0.43
	DLCO (mmol/min/kPa)	4.62	2.24	4.69	2.17	4.76	3.93	0.77
	DLCO (%N)	64.00	24.50	73.00	43.00	61.50	48.50	0.443
	6MWT (m)	287.00	95.00	289.00	230.00	323.00	286.25	0.99

**Table 2 ijms-23-15205-t002:** Patient comparison in terms of BAL parameters.

	IPAF (n = 9)		CTD-ILDs (n = 15)		IIPs (n = 16)			*p*	
	*Me*	*IQR*	*Me*	*IQR*	*Me*	*IQR*			
BAL TCC (×10^6^) N: 18.1 ± 0.9 (non-smokers); 59.8 ± 7.3 (smokers) [5]	22.3	12.4	25.05	15.35	27.05	6.59		0.8	
BAL N (%)	9.70	1.60	4.35	7.90	3.05	8.68	0.49	0.78	<0.01
N: ≤3%									
[6]									
BAL M (%) N: >85% [6]	66.30	23.5	67.30	24.90	82.95	26.48	3.23	0.19	0.03
BAL L (%)	12.80	18.00	22.90	22.48	7.75	14.00	5.28	0.07	0.09
N: 10–15%									
[6]									
BAL E (%) N: ≤1% [6]	0.90	3.85	0.45	2.35	0.85	1.40	0.95	0.62	<0.01

Abbreviations: BAL N—bronchoalveolar lavage neutrophil count; BAL M—bronchoalveolar lavage macrophage count; BAL L—bronchoalveolar lavage lymphocyte count; BAL E—bronchoalveolar lavage eosinophil count.

**Table 3 ijms-23-15205-t003:** Frequency analysis with Pearson’s χ^2^ test or Fisher–Freeman–Halton exact test for rheumatological symptoms by group membership and autoantibodies presence.

Rheumatological Symptoms	IPAF	CTD-ILDs	IIPs			
	%	%	%	χ^2^	*p*	*V*
Fever/subfebrile states	11.1	6.7	12.5	0.61	1.000	0.09
Peripheral lymphadenopathy/hepato- and splenomegaly/	11.1	6.7	6.3	0.71	1.000	0.07
Raynaud phenomenon	44.4	13.3	0	7.79	**0.008**	0.47
Arthritis/articular edema	55.6	60.0	0	14.41	**<0.001**	0.60
Articular pains	100.0	93.3	68.8	4.59	0.12	0.38
Muscle weakness	0	28.6	12.5	3.06	0.16	0.30
Sclerodactyly (proximal to elbows and knees)	0	20.0	0	3.83	0.06	0.37
Teleangiectasias	11.1	46.7	6.3	7.20	**0.02**	0.45
Sicca syndrome	0	20.0	0	3.83	0.06	0.37
Erythema (heliotrope rash/butterfly rash/DLE type symptoms/SCLE symptoms)	0	6.7	0	1.72	0.60	0.21
Morning joint stiffness > 1 h	77.8	53.3	0	17.43	**<0.001**	0.66
History of pleural effusion	11.1	6.7	0	1.94	0.51	0.20
Digital edema	22.2	33.3	0	6.51	**0.03**	0.39
**RF**						
Negative	66.7	10	14	2.35	0.338	0.24
Positive	33.3	5	2			
**Anti-CCP**						
Negative	66.7	9	15	5.36	0.060	0.36
Positive	33.3	6	1			
**ANA**						
Negative	11.1	5	13	13.25	**0.001**	0.58
Positive	88.9	10	3			

**Table 4 ijms-23-15205-t004:** Correlation between BAL protein concentrations and pulmonological test parameters.

	IL—8		TGF—β1		SPD		SPA		CXCL1	
	*Eta/r_s_*	*p*	*Eta/r_s_*	*p*	*Eta/r_s_*	*p*	*Eta/r_s_*	*p*	*Eta/r_s_*	*p*
FVC (1 −≥ 50% pred. v.; 0 −< 50% pred. v.)	0.09	0.59	0.13	0.43	**0.33**	**0.04**	0.24	0.13	0.14	0.40
DLCO (1 −≥ 60% pred.v.; 0 < 60% pred. v)	0.05	0.77	−0.09	0.60	−0.06	0.74	0.12	0.45	**−0.36**	**0.02**
FVC (l)	0.08	0.62	−0.14	0.39	−0.17	0.31	**−0.38**	**0.02**	0.15	0.38
BAL TCC (×10^6^)	**0.46**	**0.003**	0.07	0.69	0.08	0.61	0.31	0.06	**0.52**	**0.001**
BAL N (%)	−0.15	0.36	0.03	0.86	−0.16	0.33	−0.30	0.06	0.10	0.56
BAL M (%)	−0.01	0.93	−0.12	0.47	0.07	0.68	0.07	0.70	0.00	0.99
BAL L (%)	0.05	0.75	0.15	0.35	0.11	0.50	0.06	0.74	0.00	0.99
BAL E (%)	0.05	0.79	0.04	0.83	−0.18	0.27	−0.12	0.48	0.09	0.59

**Table 5 ijms-23-15205-t005:** Correlation between BAL protein concentration and autoantibodies presence.

	IL-8_mean		TGF-β1_mean		SPD_mean		SPA_mean		CXCL1_mean	
	*Eta/r_s_*	*p*	*Eta/r_s_*	*p*	*Eta/r_s_*	*p*	*Eta/r_s_*	*p*	*Eta/r_s_*	*p*
RF (1—positive; 0—negative)	0.12	0.46	0.11	0.52	0.08	0.65	0.14	0.39	0.03	0.87
anti-CCP (1—positive; 0—negative)	0.12	0.46	0.17	0.30	0.21	0.19	0.24	0.14	−0.16	0.35
ANA (1—positive; 0—negative)	0.10	0.56	**−0.35**	**0.03**	−0.12	0.46	−0.06	0.72	−0.08	0.65

**Table 6 ijms-23-15205-t006:** Regression coefficients for BAL TCC based on cytokines concentrations.

						95.0% CI for B	
	*B*	*SE*	*Beta*	*T*	*P*	*LL*	*UL*
(Constant)	21.90	7.88		2.78	0.009	5.83	37.98
IL-8_mean	0.04	0.02	0.32	2.18	0.04	0.00	0.07
TGFβ1_mean	0.00	0.00	−0.10	−0.69	0.49	0.00	0.00
SPD_mean	−0.01	0.02	−0.05	−0.30	0.77	−0.05	0.04
SPA_mean	0.00	0.00	0.17	1.09	0.28	0.00	0.01
CXCL1_mean	0.00	0.00	0.45	3.08	0.004	0.00	0.01

## Data Availability

The data presented in this study are available on request from the corresponding author. The data are not publicly available due to privacy.

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
