# Peer review of "Are There Differences in Inflammatory and Fibrotic Pathways between IPAF, CTD-ILDs, and IIPs? A Single-Center Pilot Study"

_ijms, 2022, doi:10.3390/ijms232315205_

Round 1
Reviewer 1 Report
1. These authors have reported information from a single center pilot study evaluating possible inflammatory and fibrotic pathways in patients in distinct groups of interstitial lung disease. In particular, they measured transforming growth factor beta, surfactant proteins A and D, interleukin-8, and chemokine 1 in bronchoalveolar lavage fluid from these patients. They found that interleukin-8 and transforming growth factor beta 1 levels were lower in patients with IPF than in CTD-ILD patients.
2. Introduction: The introduction seems somewhat long. The authors discussed several proteins which were not measured in this study.
3. Methods: In the section on serologic testing, there is no information regarding rheumatoid factor measurements. In addition, HRCT evaluation is in that paragraph and probably should be in a separate paragraph. Also, the method used to evaluate the CT scans is not provided. The section on protein concentration measurements includes information about neutrophil activating proteins 3 which is not reported in this study.
4. Results: In table 1, the information on FEV1 [percent] is reported as a decimal fraction. For other tests the percent predicted is reported as a whole number. The sentence regarding radiological pattern of usual interstitial pneumonia probably needs more information as to the exact pattern used to classify patients. It might be helpful to add a representative HRCT for each group. Table 2 has information about the percent neutrophils, macrophages, lymphocytes, and eosinophils in lavage fluid. The information provided as the median and interquartile range. However, the information in columns 7 and 8 is not explained. The authors state that table 3 contains information regarding autoantibodies and titers. However, that information does not appear to be in this table. The authors report the number of patients who tested positive for rheumatoid factor. In the next paragraph, they report patients who tested negative for ANA. This is potentially confusing unless the reader is careful. That information might better be reported as the number tested positive for ANA.
5. Discussion: Do the authors have any information regarding the duration of disease in the patients in these 3 groups? This may have an effect or association with the concentration of various biomarkers in lavage fluid. In the discussion the authors comment several times that the small sample size limits conclusions. It seems to me that these comments could be summarized in a single paragraph on limitations. Do the authors plan to do serial studies in patients? This might be more informative than a single measurement in a patient in whom the time course for disease progression is unknown.
Author Response
Dear Reviewer,
Thank you for your valuable comments which contributed to improvement of this manuscript. Below, we present alterations made to the manuscript according to your comments. We used “track changes” mode to make the modifications visible.
Ad 2. We shortened the introduction and removed references to biomarkers which we did not measure in our study, please see section “Introduction”.
Ad 3. We added a reference to method of rheumatoid factor concentration assessment, please see section “Serological testing”. We added a separate section “Radiological imaging” where we make reference to high-resolution computed tomography protocol used in our clinic. Thank you for bringing to your attention the mention of NAP3, which was simply a misprint.
Ad 4. We corrected the FEV1 information and presented it as percentage of predicted value (please see Table 1). We added a representative HRCT scan for each study group, please see Figures 1, 2 and 3.
We also made clear which guidelines we used to diagnose usual interstitial pneumonia .
Ad 5. We have information on symptoms duration in each individual in the study. We included the relationships between symptom duration and BAL biomarker concentration in each study group, however, no statistical significance was revealed. Neither was there any statistical significance in symptom duration between the three study groups. These remarks are included in the “Results” section “3.5. Relationship between BAL protein concentration and symptom duration”.
We included a “Study limitations” section to avoid repeated remarks on sample size throughout the discussion. We added information on planned serial testing to reveal potential relationships with patient-related outcomes.
Thank you for your valuable time.
Yours sincerely,
Patrycja Rzepka-Wrona, Szymon Skoczyński, Adam Barczyk

Reviewer 2 Report
This is a nice manuscript with study protocol and patient selection. Studying BAL fluid biomarkers and its correlations with clinic and radiologic findings in ILD is a challenging research area for pulmonologists.
Author Response
Dear Reviewer,
Thank you for your valuable time and your kind comments.
Yours sincerely,
Patrycja Rzepka-Wrona, Szymon Skoczyński, Adam Barczyk